# Antileukemic Activity of Twig Components of Caucasian Beech in Turkey

**DOI:** 10.3390/molecules24213850

**Published:** 2019-10-25

**Authors:** Wataru Shida, Hiroshi Tateishi, Yurika Tahara, Mikako Fujita, Doaa Husham Majeed Alsaadi, Masato Watanabe, Ryoko Koga, Mohamed O. Radwan, Halil I. Ciftci, Sevgi Gezici, Yuki Kurauchi, Hiroshi Katsuki, Masami Otsuka, Koji Sugimura, Mikiyo Wada, Nazim Sekeroglu, Takashi Watanabe

**Affiliations:** 1Medicinal and Biological Chemistry Science Farm Joint Research Laboratory, Faculty of Life Sciences, Kumamoto University, 5-1 Oe-honmachi, Chuo-ku, Kumamoto 862-0973, Japan; wtrsd060725@gmail.com (W.S.); htateishi@kumamoto-u.ac.jp (H.T.); 196y2011@st.kumamoto-u.ac.jp (Y.T.); kk1205@kumamoto-u.ac.jp (R.K.); mohamedradwan@kumamoto-u.ac.jp (M.O.R.); hiciftci@kumamoto-u.ac.jp (H.I.C.); motsuka@gpo.kumamoto-u.ac.jp (M.O.); 2Department of Medicinal Plant, Faculty of Life Sciences, Kumamoto University, 5-1 Oe-honmachi, Chuo-ku, Kumamoto 862-0973, Japan; konkondooo@yahoo.com (D.H.M.A.); wata-m@gpo.kumamoto-u.ac.jp (M.W.); sugimura@kumamoto-u.ac.jp (K.S.); 3Department of Drug Discovery, Science Farm Ltd., 1-7-30-805 Kuhonji, Chuo-ku, Kumamoto 862-0976, Japan; 4Chemistry of Natural Compounds Department, Pharmaceutical and Drug Industries Research Division, National Research Centre, Dokki 12622, Cairo, Egypt; 5Advanced Technology Application and Research Center, Kilis 7 Aralik University, Kilis 79000, Turkey; sevgigezici@kilis.edu.tr; 6Department of Molecular Biology and Genetics, Faculty of Arts and Sciences, Kilis 7 Aralik University, Kilis 79000, Turkey; 7Department of Chemico-Pharmacological Sciences, Graduate School of Pharmaceutical Sciences, Kumamoto University, 5-1 Oe-honmachi, Chuo-ku, Kumamoto 862-0973, Japan; kurauchy@kumamoto-u.ac.jp (Y.K.); hkatsuki@gpo.kumamoto-u.ac.jp (H.K.); 8Department of Instrumental Analysis, Faculty of Life Sciences, Kumamoto University, 5-1 Oe-honmachi, Chuo-ku, Kumamoto 862-0973, Japan; wadayo@kumamoto-u.ac.jp; 9Department of Horticulture, Faculty of Agriculture, Kilis 7 Aralik University, Kilis 79000, Turkey; nsekeroglu@gmail.com

**Keywords:** Caucasian beech, beech, twig, Turkey, anti-leukemic activity

## Abstract

Despite the development of a range of anti-cancer agents, cancer diagnoses are still increasing in number, remaining a leading cause of death. Anticancer drug treatment is particularly important for leukemia. We screened Turkish plants and found the unique antileukemic activity of twig components in Turkish Caucasian beech, selectively inducing apoptosis in leukemia cells. This effect is unique among some kinds of beeches, presumably related to oxidative stress. This study would lead to effective use of discarded material, i.e., twig of beech, and a new anti-leukemic drug based on large tree.

## 1. Introduction

Despite the development of a range of anti-cancer agents, including molecular target drugs and antibody drugs, cancer diagnoses are still increasing in number, remaining a leading cause of death [1]. Anticancer drug treatment is particularly important for leukemia, in addition to hematopoietic stem cell transplantation, since surgical resection is not applicable to hematological tumors. The number of new cases of leukemia and deaths due to leukemia was 467,000 and 310,000 in 2016, respectively [2].

As antileukemia drugs, antimetabolites, anthracycline antibiotics, alkylating agents, plant alkaloids, steroids, and molecular target drugs have been developed and clinically used. We reported that pentacyclic triterpene of a plant origin, namely gypsogenin, and related synthetic derivatives, have an anti-leukemic activity, efficiently inhibiting Bcr-Abl kinase [3,4]. In this study, we focused on plants in Turkey, since the country has a characteristic geography and the climate of the Europe-Asia interface area is surrounded by Mediterranean Sea and Black Sea. We screened Turkish plants and found the unique antileukemic activity of twig components in Turkish Caucasian beech, which is evidently different from Japanese beech trees.

## 2. Results and Discussion

In September 2016, wild plants, market plants, and farm plants were collected in Turkey. These materials were dried and extracted with 70% EtOH and the solvent was evaporated. The residues were dissolved in DMSO and a Turkish plant library consisting of 164 samples was thus obtained. We employed leukemic cell lines to screen this library.

Two different leukemic cells, namely, the human acute monocytic leukemia cell line, THP-1, and the human chronic myelogenous leukemia cell line, K562, were used for the first screening. Each sample solution (100 μg/mL) was added to the cells. After being incubated for 3 days, a MTT assay was performed to determine cellular viability. Five samples (A–E) showing relative viability lower than 30% in at least one of the two cells were selected (Figure 1a).

The toxicities of these samples was next examined using peripheral blood mononuclear cells (PBMC) from healthy donors. The PBMC were incubated with samples A–E (100 μg/mL) for 3 days, and cellular viability was examined. As shown in Figure 1b, samples C–E showed toxicity against normal cells. In contrast, A and B were non-toxic to normal cells and selectively toxic to leukemic cells. Sample A was extracted from the twigs of Caucasian beech (*Fagus orientalis* LIPSKY) (Figure 1c) in northeast part of Turkey facing the Black Sea. Sample A showed cytotoxic activity against other leukemia cell lines, namely, Jurkat and MT-2, and was as potent as THP-1 and K562 (Figure 1d). Sample B was extracted from flowers of *Colchicum speciosum* Steven. This plant has been already been extensively investigated [5,6], and its extract is known to have anticancer activities [7,8]. Samples C, D, and E were extracts of Vitex angus-castus (seed), Boswellia sp. (resin), and *Ferula meifolia* (Fenzl) Boiss. (stem bark), respectively.

We focused on sample A. Then, we examined whether this anti-leukemic effect was evidence of the induction of apoptosis or not. Thus, THP-1 cells were treated with sample A for 6 h and the cells were washed, stained with Hoechst 33342, Annexin V-FITC, and EtD-III, and observed by a fluorescence microscope. Here, Hoechst 33342 stained the nuclei of the cells. The representative microscopic field is shown in Figure 2a. Some cells were stained only with Annexin V (colored green), meaning apoptosis, and two cells were stained with both Annexin V and EtD-III (colored red), indicating late apoptosis or necrosis. Furthermore, immunoblot analysis using the anti-Caspase-3 antibody was conducted at the same time after treatment of THP-1 cells with sample A. As shown in Figure 2b, the cleavage of Caspase-3 was observed. These results show sample A induces apoptosis in the leukemia cells.

Next, we checked the cytotoxic effect of samples from twigs of Siebold’s beech (*Fagus crenata*, BLUME) and Inu beech (Japanese beech) (*Fagus japonica*, MAXIM), both collected in Japan. The 70% EtOH extract was prepared as described, and the cytotoxicity against THP-1 and K562 cell lines was examined. As shown in Figure 3a, the samples from these beeches in Japan did not show cytotoxicity, unlike that from Caucasian beech.

To explore the differences among these beech samples, cell signaling related to apoptosis was examined. In fact, various plant products are known to regulate cell signaling [9,10,11]. First, nuclear factor-κB (NF-κB) activation [12,13] was observed by fluorescence microscope. Here, the HeLa cell line was used in this experiment, since a leukemia cell has a big nucleus, and it is difficult to observe nuclear localization of proteins using these cells. The HeLa cell line was incubated with each beech sample, and the localization of NF-κB p65 was examined. As shown in Figure 3b, a small amount of NF-κB and p65, normally localized in cytoplasm, was observed to translocate to the nucleus in the presence of three beech samples. Furthermore, immunoblotting of the same samples (Figure 3c) showed the degradation of some amount of IκBα, supporting the ability of the all beech samples to activate NF-κB. However, this activity does not explain the apoptosis induced only by the Caucasian beech sample.

We then conducted the same fluorescence microscopic observation to examine the nuclear localization of nuclear factor erythroid 2-related factor 2 (Nrf2), which is known to be activated in response to oxidative stress [14,15]. As shown in Figure 3d, big changes were not observed in the control and with samples of Siebold’s beech and Inu beech. In contrast, some amounts of Nrf2 protein were observed to enter the nucleus in the presence of Caucasian beech sample. Usually, the activation of Nrf2 to enter the nucleus protects cells from apoptosis. In this case, there is a possibility that the Caucasian beech sample gave oxidative stress, surmounting the threshold for protection by Nrf2.

Taken together, we found that twig components in Turkish Caucasian beech selectively induce apoptosis in leukemia cells. This effect is unique among some kinds of beeches. People have mainly used grasses and shrubs as medicinal plants in this regard. It was found that yew trees contain anticancer paclitaxel [16] and willow produces the analgesic salicin, which is a lead of aspirin [17]. However, the application of large trees for medicine has been limited. Whereas the stems of beech have been used as wood materials for furniture etc., twigs have been discarded. This study shows the possibility of effective use of discarded materials. The extracts used herein are a mixture of beech products, and we will isolate chemicals having the anti-leukemic activity and reveal the detailed mechanism of action in the near future.

## 3. Method and Materials

### 3.1. Sample Collection and Extraction

In September 2016, plant materials were collected from wild plants, market plants, and farm plants in Turkey. These wild plants were in northeast part of Turkey, facing the Black Sea. The materials contained the roots, stems, twigs, leaves, fruit, shells and seeds of plants. These materials were cleaned and dried at 50 °C for 7–10 days with ADVANTEC (Tokyo, Japan) DRG400AA. The dried samples were cut and/or grinded and soaked into 70% EtOH (20 times volume per plant weight). Then the liquid with each sample was sonicated with sonicator Ultrasonic Cleaner (Dual Frequency, ASU-D Series) (AS ONE, Osaka, Japan) at 60 °C for 2 h. The solvent was evaporated, and the residue was dried under vacuum overnight. The dried sample was then dissolved in DMSO (10 mg/mL) to prepare a Turkish plant library consisting of 164 samples. Twigs of Siebold’s beech and Inu beech were collected from wild plants in Kumamoto, Japan, and these 70% EtOH extracts were prepared as described above.

### 3.2. Cells

The human acute monocytic leukemia cell line THP-1 [18] was cultured in RPMI-1640 (Wako Pure Chemical, Osaka, Japan) medium supplemented with 10% heat-inactivated fetal bovine serum (FBS) (Sigma-Aldrich, St. Louis, MO, USA) and 55 μM 2-mercaptoethanol (Thermo Fisher Scientific, Waltham, MA, USA). The human chronic myelogenous leukemia cell line K562 [19], human acute lymphocytic leukemia cell line Jurkat [20], and the human T cell line MT-2, infected with the human T-cell leukemia virus [21], were culture in RPMI-1640 (Wako Pure Chemical) medium supplemented with 10% heat-inactivated FBS (Sigma-Aldrich). The human cervical cancer cell line HeLa [22] was maintained in Dulbecco’s modified Eagle’s medium (Sigma-Aldrich) supplemented with 5% FBS (Sigma-Aldrich). Normal human peripheral blood mononuclear cells (PBMC, Precision for Medicine, Bethesda, MD, USA) were purchased and cultured in a RPMI-1640 (Wako Pure Chemical) medium supplemented with 10% heat-inactivated FBS (Sigma-Aldrich).

### 3.3. MTT Assay

The cells were incubated with DMSO (Wako Pure Chemical) or DMSO solution of samples (1/100 volume) for 3 days. Then, MTT assay was performed as previously described [23].

### 3.4. Fluorescence Microscopy and Immunoblot Analysis

The cells were incubated with DMSO (Wako Pure Chemical) or a DMSO solution of sample A (extract of twig of Caucasian beech, 100 μg/mL) (1/100 volume) for 4 h. Then, the cells were washed and stained with Hoechst 33342 (Thermo Fisher Scientific), Annexin V-FITC (PromoKine, Heidelberg, Germany), and EtD-III (PromoKine), as previously described [23,24]. For immunostaining, the NFκB p65 antibody (Santa Cruz, Dallas, TX, USA) and anti-Nrf2 antibody (Abcam, Cambridge, UK) were used, and the staining was basically processed according to previously reported method [25]. The Leica (Wetzlar, Germany) TCS SP5 fluorescence microscope was used. For immunoblot analysis, THP-1 cells were incubated with or without sample A for 4 h, as described above. The cells were then lyzed and analyzed by immunoblotting, as previously described [26,27,28]. As an antibody, anti-Caspase-3 (Cell signaling technology, Danvers, MA, USA), IκB-α antibody (Cell signaling technology), or anti-β-actin (Sigma-Aldrich) was used.

## Figures and Tables

**Figure 1 molecules-24-03850-f001:**
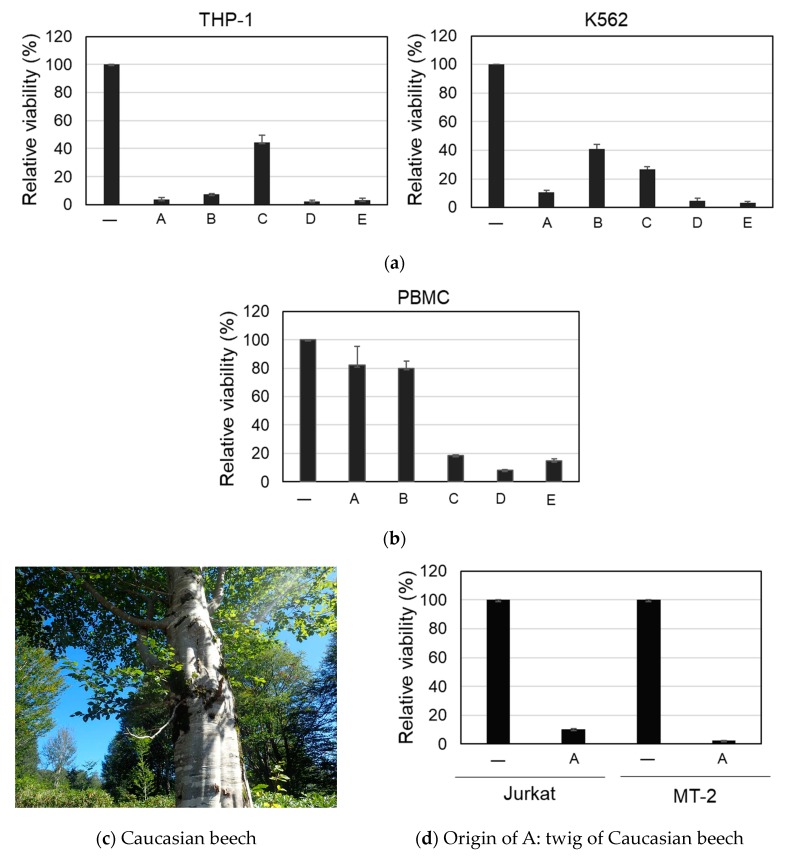
Cytotoxic effect of 70% EtOH extract from plant materials. The cells were incubated with each sample (100 μg/mL) for 3 days, and a MTT assay was performed. The relative viability is shown. (**a**) Viability of THP-1 and K562 cells treated with the samples A–E in the first screening. (**b**) Viability of peripheral blood mononuclear cells (PBMC) treated with the samples A–E in the second screening. (**c**) Photograph of Caucasian beech in Turkey. (**d**) Viability of Jurkat and MT-2 cells treated with sample A.

**Figure 2 molecules-24-03850-f002:**
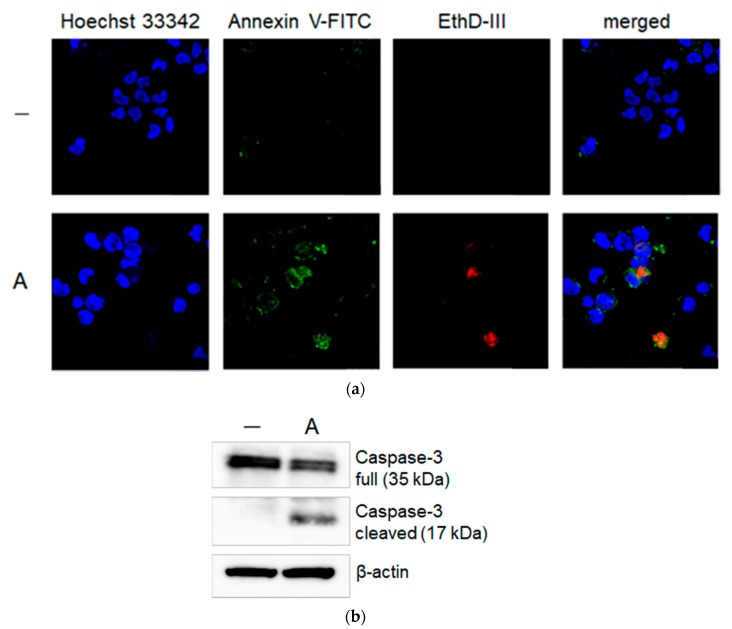
Apoptotic effect of sample A on THP-1 cells. The cells were incubated with sample A (100 μg/mL) for 6 h and analyzed. (**a**) Fluorescence microscopic observation after staining with Hoechst 33342, Annexin V-FITC, and EthD-III. (**b**) Immunoblot analysis using anti-Caspase-3 and anti-β-actin antibodies.

**Figure 3 molecules-24-03850-f003:**
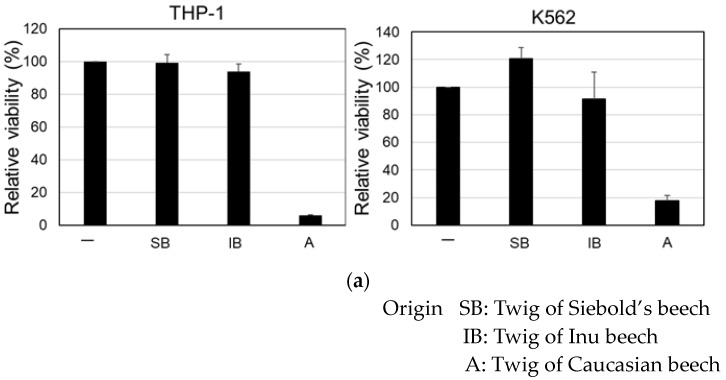
Cytotoxic effect of 70% EtOH extract from twigs of Siebold’s beech (SB), Inu beech (IB) and Caucasian beech (A). The cells were incubated with each sample (100 μg/mL). (**a**) MTT assay at 3 d post-addition of the samples to THP-1 and K562 cells. The relative viability is shown. (**b**) Fluorescence microscopic observation after staining with Hoechst 33342 and anti-NF-κB p65 at 4 h post-addition of the samples to HeLa cells. (**c**) Immunoblot analysis using anti-IκBα and anti-β-actin antibodies at 4 h post-addition of the samples to HeLa and THP-1 cells. (**d**) Fluorescence microscopic observation after staining with Hoechst 33342 and anti-Nrf2 at 4 h post-addition of the samples to HeLa cells. SB: Sample of twig of Siebold’s beech. IB: Sample of twig of Inu beech.

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
