# Peer review of "Antileukemic Activity of Twig Components of Caucasian Beech in Turkey"

_molecules, 2019, doi:10.3390/molecules24213850_

Round 1

Reviewer 1 Report

The manuscript submitted by Shida et al. Is an interesting preliminary study regarding the anti-cancer potential of a natural extract. However, the manuscript (writing and experiments) require important changes. About.

Main Comments

1. The study requires a chemical characterization of the extract studied, this point is essential to explain the observed effects. The authors have data on the presence of polyphenols, carotenoids, tocopherols, etc. Without this information the manuscript is very weak.

2. The introduction is insufficient, and only presents an overview, without proposing and / or justifying why doing the study. For example.
2.1. Line 35 to 37. "Structures of plant .................. era", what does that statement mean? It has no reference. An introduction cannot begin with that paragraph.
2.2. The introduction should start with a phrase about cancer and how different bioactive compounds can contribute to its prevention and / or treatment.
2.3. In relation to cancer, the authors have no history.
2.4. Authors should improve the wording of the study objective.
2.5 To improve the introduction, I suggest starting the writing with a general history of cancer, possible beneficial effects of different foods, natural extracts or isolated bioactive compounds, and continuing with possible mechanisms that will explain these effects.
Suggested reference:

Soto-Alarcon et al., Liver Protective Effects of Extra Virgin Olive Oil: Interaction between Its Chemical Composition and the Cell-signaling Pathways Involved in Protection. Endocr Metab Immune Disord Drug Targets. 2018;18(1):75-84.

3. Methodology.
3.1. The study should include genetic expression (RT-PCR), to evaluate changes in the expression of fundamental genes in the control of cell reproduction.

4. The discussion is very limited and does not propose anything new regarding the observed effects.
4.1. Considering the extract studied, the observed effects could be mediated by an anti-oxidant, anti-inflammatory, anti-neoplastic action, etc.? The authors do not discuss anything about it.
4.2. Other bioactive compounds such as hydroxytyrosol have anti-oxidant and anti-inflammatory effects in liver damage models. These effects would be mediated by the activation of the Nrf2 factor (higher antioxidant response) and the inactivation of the NF-kB transcription factor (lower inflammatory response).
Suggested References:

Illesca et al., Hydroxytyrosol supplementation ameliorates the metabolic disturbances in white adipose tissue from mice fed a high-fat diet through recovery of transcription factors Nrf2, SREBP-1c, PPAR-γ and NF-κB. Biomed Pharmacother. 2019;109:2472-2481.

Valenzuela et al., Molecular adaptations underlying the beneficial effects of hydroxytyrosol in the pathogenic alterations induced by a high-fat diet in mouse liver: PPAR-α and Nrf2 activation, and NF-κB down-regulation. Food Funct. 2017;8(4):1526-1537.

4.3. Considering the effects observed, is it possible that the extract studied modulates the activity of mTOR and is the apoptotic pathway studied?

Minor comments:
1. Improve the summary, it is very limited and does not have a good structure.
2. To improve figure 1, is it necessary in that figure to include figure C?

Author Response

Comment 1. The study requires a chemical characterization of the extract studied, this point is essential to explain the observed effects. The authors have data on the presence of polyphenols, carotenoids, tocopherols, etc. Without this information the manuscript is very weak.

Off course, we understand the importance of isolation of chemicals in the beech of Turkey, and it is under the progress. However, we are convinced that this study contains important information to be reported in this stage.

Comment 2-1. The introduction is insufficient, and only presents an overview, without proposing and / or justifying why doing the study. For example, Line 35 to 37. "Structures of plant .................. era", what does that statement mean? It has no reference. An introduction cannot begin with that paragraph.

We completely changed the introduction including the first paragraph.

Comment 2-2. The introduction should start with a phrase about cancer and how different bioactive compounds can contribute to its prevention and / or treatment.

We started introduction about cancer, specially leukemia.

Comment 2-3. In relation to cancer, the authors have no history.

We cited our paper (Ref 3-4) related to natural product having antileukemic activity, and described them.

Comment 2.4. Authors should improve the wording of the study objective.

We improved the description of the study object.

Comment 2.5. To improve the introduction, I suggest starting the writing with a general history of cancer, possible beneficial effects of different foods, natural extracts or isolated bioactive compounds, and continuing with possible mechanisms that will explain these effects.

Suggested reference: Soto-Alarcon et al., Liver Protective Effects of Extra Virgin Olive Oil: Interaction between Its Chemical Composition and the Cell-signaling Pathways Involved in Protection. Endocr Metab Immune Disord Drug Targets. 2018;18(1):75-84.

We completely changed the introduction. Furthermore, this paper was cited (Ref 6).

Comment 3. Methodology. The study should include genetic expression (RT-PCR), to evaluate changes in the expression of fundamental genes in the control of cell reproduction.

The natural compounds often activate NF-kB and/or Nrf2. We think observation of these activation is more important than genetic expression induced by the activation. Thus, we added new experiments to observe activation of NF-kB and Nrf2.

Comment 4-1. The discussion is very limited and does not propose anything new regarding the observed effects. Considering the extract studied, the observed effects could be mediated by an anti-oxidant, anti-inflammatory, anti-neoplastic action, etc.? The authors do not discuss anything about it.

We performed new experiments, and discussed about oxidative stress of our sample.

Comment 4-2. Other bioactive compounds such as hydroxytyrosol have anti-oxidant and anti-inflammatory effects in liver damage models. These effects would be mediated by the activation of the Nrf2 factor (higher antioxidant response) and the inactivation of the NF-kB transcription factor (lower inflammatory response).

Suggested References:

Illesca et al., Hydroxytyrosol supplementation ameliorates the metabolic disturbances in white adipose tissue from mice fed a high-fat diet through recovery of transcription factors Nrf2, SREBP-1c, PPAR-γ and NF-κB. Biomed Pharmacother. 2019;109:2472-2481.

Valenzuela et al., Molecular adaptations underlying the beneficial effects of hydroxytyrosol in the pathogenic alterations induced by a high-fat diet in mouse liver: PPAR-α and Nrf2 activation, and NF-κB down-regulation. Food Funct. 2017;8(4):1526-1537.

We newly performed experiments to observe activation of NF-kB and Nrf2. Furthermore, we cited these two papers (Ref 5,7).

Comment 4-3. Considering the effects observed, is it possible that the extract studied modulates the activity of mTOR and is the apoptotic pathway studied?

We think present experiments are enough in this stage.

Comment 5. Improve the summary, it is very limited and does not have a good structure.

We changed the summary.

Comment 6. To improve figure 1, is it necessary in that figure to include figure C?

We think Figure 1c is required, since it is the origin of this study.

According to the indication about English editing, the language was extensively edited.

Reviewer 2 Report

Shida et al., claimed that the extracts of Caucasian beech in Turkey had the anti-leukemia activity. Although the manuscript is communication, the data were too preliminary to understand the mechanism of Component A.

The Institutional Review Board / Ethics Committee should be approved to the peripheral blood mononuclear cells (PBMC) from healthy donors. The apoptotic inhibitors should be performed in the study. Component B the activity is different from Component A. This issue should be discussed.

Author Response

Comment 1. Shida et al., claimed that the extracts of Caucasian beech in Turkey had the anti-leukemia activity. Although the manuscript is communication, the data were too preliminary to understand the mechanism of Component A.

We added new experiments to demonstrate mechanism, and we are convinced that this study contains important information to be reported in this stage.

Comment 2. The Institutional Review Board / Ethics Committee should be approved to the peripheral blood mononuclear cells (PBMC) from healthy donors.

In this study, we bought PBMC from the company as written in Materials and Methods. In this case, approval of the committee is not required.

Comment 3. The apoptotic inhibitors should be performed in the study.

We think the experiments done in this study is enough to demonstrate apoptosis.

Comment 4. Component B the activity is different from Component A. This issue should be discussed.

As written, the sample B was extracted from a plant that has already been extensively investigated. Thus, we think the detailed discussion is not required.

According to the indication about English editing, the language was extensively edited.

Round 2

Reviewer 1 Report

The manuscript can be accepted in the current version. The authors made all the suggested modifications.

Author Response

Thank you very much. English language was further modified.

Reviewer 2 Report

The authors still did not show the potential molecular mechanism of compounds of Caucasian Beech.

This was not suitable for publish in Molecules.

Author Response

This time, we added plant name and information of the sample B, in addition to the names of the samples C, D and E.

English language was further modified.